# Monochorionic Diamniotic Twins with Sex Discordance: Case Series

**DOI:** 10.3390/diagnostics15030372

**Published:** 2025-02-04

**Authors:** Valentina Sala, Luigina Spaccini, Stefano Faiola, Daniela Casati, Arianna Laoreti, Lisanne S. A. Tollenaar, Enrico Lopriore, Mariano M. Lanna

**Affiliations:** 1Fetal Therapy Unit “U. Nicolini”, Vittore Buzzi Children’s Hospital, University of Milan, 20154 Milan, Italy; valentina-sala@live.it (V.S.); daniela.casati@asst-fbf-sacco.it (D.C.); arianna.laoreti@asst-fbf-sacco.it (A.L.); 2Vittore Buzzi Children’s Hospital Clinical Genetic Service, University of Milan, 20154 Milan, Italy; luigina.spaccini@asst-fbf-sacco.it; 3Division of Fetal Medicine, Department of Obstetrics, Leiden University Medical Center, 2333 ZA Leiden, The Netherlands; 4Division of Neonatology, Department of Pediatrics, Leiden University Medical Center, 2333 ZA Leiden, The Netherlands; e.lopriore@lumc.nl

**Keywords:** twin discordance, zygosity, sesquizygotic, chimerism

## Abstract

**Background and Clinical Significance:** Ultrasonographic diagnosis of twin pregnancies has become routine, with chorionicity playing a crucial role in assessing associated risks. Traditionally, monochorionic (MC) twins were believed to derive from a single zygote, ensuring sex concordance. However, recent cases of dizygotic monochorionic (MCDZ) twins challenge this paradigm. In this paper, four cases of MCDZ twins with sex discordance are described. **Case presentation:** Case 1 and case 2 describe two spontaneous MC/diamniotic twin pregnancies in which sex discordance between twins was attributed to blood chimerism. Case 3 is about a MC/diamniotic twin pregnancy derived from a single blastocyst transfer after in vitro fertilization (IVF), and that was complicated by twin-to-twin transfusion syndrome, with zygosity testing confirming the dizygosity. Case 4 is a twin anemia polycythemia sequence diagnosed after birth in twins considered dichorionic during pregnancy (due to sex difference) and defined as monochorionic after placental examination. **Conclusions:** The prevalence of monochorionic dizygotic (MCDZ) twins remains uncertain, and many cases likely go unnoticed, particularly when twins are of the same sex. In twin pregnancies, determining chorionicity during the first-trimester ultrasound (US) is critical. Accurate identification of monochorionicity is essential for managing potential complications. Careful verification of sex concordance between twins is necessary. In cases of sex discordance, amniocentesis is required for karyotype evaluation and zygosity testing.

## 1. Introduction

The characterization of a twin pregnancy is closely linked to determining its zygosity (the genetic similarity between twins) and chorionicity (type of placentation) via ultrasonography (US) during the first trimester [1]. Chorionicity is classified as dichorionic (DC) or monochorionic (MC), to assess a stratification of risk with specific complication arising from placental anastomosis in MC [2]. Zygosity refers to the genetic similarity between twins and is broadly categorized as monozygotic (MZ) or dizygotic (DZ).

Monozygotic (MZ) twins: These twins originate from a single fertilized egg (zygote) that splits into two embryos. MZ twins share 100% of their genetic material and are often referred to as “identical twins”, although environmental factors can cause minor differences in appearance.Dizygotic (DZ) twins: These twins result from the fertilization of two separate eggs by two different sperm. Often termed “fraternal twins”, DZ twins share approximately 50% of their genetic material, similar to typical siblings.

Determining zygosity is clinically significant, as it predicts potential complications during pregnancy and informs postnatal care.

Chorionicity, the type of placental relationship between twins, is classified as either dichorionic (DC) or monochorionic (MC):1.Dichorionic (DC) twins: Each twin has a separate placenta and amniotic sac, associated with lower risks of complications due to absence of shared placental blood vessels
o DC twins can be either DZ or MZ. DZ pregnancies naturally lead to separate placentas and sacs. In MZ pregnancies, early splitting of the single zygote (within three days post-fertilization) results in dichorionic placentation.o Sex discordance (one male and one female twin) is commonly observed in DC twins, particularly in DZ pregnancies.2.Monochorionic (MC) twins: These twins share the same placenta and, in some cases, a single amniotic sac (monochorionic–monoamniotic, or MCMA). MC twins are almost always MZ because they arise from the splitting of a single zygote after the chorion has already formed (between days 4 and 8 post-fertilization).
o Shared placentation increases the risk of complications, such as TTTS, twin anemia–polycythemia sequence (TAPS), and selective fetal growth restriction (sFGR).o Sex concordance (both twins being the same gender) is traditionally expected in MC twins, assuming they are monozygotic.


For many years, it was believed that all MC twins were monozygotic. However, rare cases of dizygotic monochorionic (MCDZ) twins—genetically dissimilar twins sharing a placenta—have challenged this notion. Monozygotic (MZ) twins are genetically identical and dizygotic twins share 50% of their DNA identity. In DC twins, sex discordance is reported in half of cases, since they derive from two zygotes in almost two-thirds of cases [3]; with MC twins, assuming that they derive from a single zygote, there should always be a sex concordance [4]. However, in the early 2000s, a dizygotic monochorionic (MCDZ) twin pregnancy case was reported [5], giving rise to studies related to various phenomena like chimerism or dizygosity in MC twins [6] and calling into question the long-standing hypothesis that MC twins are MZ. Here, we report four cases of MC twins with sex discordance due to varying mechanisms.

## 2. Case Report

### 2.1. Case 1

A quartigravida, 39-year-old with a spontaneous MC/diamniotic (DA) twin pregnancy was referred to our unit for US surveillance in 2006. A chorionic villus sampling (CVS) for maternal choice was performed prior to referral, and a normal male karyotype (46, XY) was identified. Biweekly US monitoring starting from 16 weeks of gestation did not reveal any complication until 24 weeks, when a sex discordance was suspected. Amniocentesis of the discordant twin revealed a female karyotype. Cord blood analysis after delivery at 36 weeks of gestation confirmed discordant sexual chromosomes with a chimerism: 46, XX (33)/46, XY (17) and 46, XY (34)/46, XX (13) out of 50 metaphases analyzed. Placental examination demonstrated monochorionicity, and no abnormalities were reported on external and internal genitalia. Follow-up did not reveal any childhood complication until 5 years of age.

### 2.2. Case 2

A quartigravida, 31-year-old patient with a spontaneous MC/DA pregnancy underwent a first-trimester screening test for trisomy 21 in 2018, with a low risk calculated for both twins. The diagnosis of a selective fetal growth restriction (sFGR) at 18 weeks was followed by a weekly US until delivery at 34 weeks due to the onset of spontaneous labor. At birth, due to unexpected sex discordance between the newborns, blood samples were collected for karyotype determination, revealing a 98.5% male karyotype (46, XY) and a 1.5% female karyotype (46, XX) in the boy and a normal female karyotype (46, XX) in 48% of cells and a normal male karyotype (46, XY) in 52% of the cells in the girl. After genetic counseling, buccal tissue sampling was performed in both twins to extract the DNA for analysis of polymorphic markers with no evidence of chimerism in peripheral tissue. In the meantime, histological analysis of the placenta was performed to confirm monochorionicity. At the time of this report, the follow-up, conducted at 9 months of age, resulted in normal findings, with no genital abnormalities.

### 2.3. Case 3

A 37-year-old primigravida patient was referred at our unit at 15 weeks of pregnancy for suspected twin-to-twin-transfusion syndrome (TTTS) in a MC/DA twin pregnancy derived from a single blastocyst transferred after ICSI (Intra-Cytoplasmic Sperm Injection). The first-trimester screening test showed a nuchal translucency of 4.1 mm and 1.1 mm with a high risk for trisomy 21 (1:13) and 18 (1:34); these findings were not confirmed by cell-free fetal DNA. US revealed a discordance in fetal sex, without an oligo-polyhydramnios sequence. To assess if it was a case of ambiguous genitalia, amniocentesis was performed on both twins for classical and molecular karyotype analysis, along with a zygosity test, and a normal karyotype was identified for both the male and the female twin. A zygosity test using polymorphic alleles segregation analysis (on fetal DNA extracted from cultured amniotic cells of both twins) confirmed that the two twins were dizygotic. After two weeks of gestation, twins developed a stage I TTTS, and due to the presence of a cardiac overload in the recipient twin, fetoscopic laser surgery was performed at 18.5 weeks. One week after the procedure, a chorioamniotic detachment was observed, with resolution of TTTS. Weekly follow-up was uneventful, with a resolution of cardiac overloads in the former recipient twin, but at 27 weeks, a complete septostomy, which led to cord entanglement between twins (referred to as iatrogenic monoamnioticity), was diagnosed. The rupture of interamniotic membrane resulted in amniotic limb syndrome on the left arm of the female twin, the former recipient. Delivery was planned with a cesarean section at 30 weeks to avoid further deterioration as a consequence of the amniotic band syndrome. At birth, amniotic bands were wrapped around the left forearm, causing constriction, and were treated with a volar fasciotomy of the forearm extending to the hand, including the opening of the carpal tunnel. Placental examination with colored dye injection showed the absence of residual anastomoses (Figure 1). At histological evaluation, the inter-twin membrane confirmed the twin to be MCDA. The physical examination showed no genital abnormalities.

### 2.4. Case 4

A 32-year secundigravida patient was diagnosed with an MCDA pregnancy at a gestational age of 9 weeks. At 14 weeks of gestation, uncertainty arose regarding the chorionicity of the pregnancy, as the inter-twin membrane showed both a lambda sign (suggestive of a DC/diamniotic (DA) twin) and a T-sign (suggestive of an MCDA twin), depending on the sonographic view. As the sexes of the twins were different, the pregnancy was classified as a dichorionic (DC). The pregnancy progressed uneventfully. At 37 weeks of gestation, the membranes ruptured, followed by an uncomplicated vaginal delivery. At birth, twin 1 (boy) weighed 2775 g and was plethoric, while twin 2 (girl) weighed 2410 g and was pale. A full blood count revealed a hemoglobin (Hb) value of 23.2 g/dL, and a reticulocyte count ratio of 40 promille for twin 1, and a Hb value of 13.8 g/dL and reticulocyte count of 60 promille for twin 2. The inter-twin Hb difference was 9.4 g/dL, and the reticulocyte count ratio was 1.5, raising suspicion of a twin anemia polycythemia sequence (TAPS). To confirm the diagnosis and reevaluate chorionicity, the placenta was first injected with color dye and then sent for microscopic examination. Dye injection of the placental vessels revealed numerous minuscule vascular anastomoses: 27 arterio-venous and 15 veno-arterial anastomoses, without arterio-arterial anastomoses (Figure 2). The maternal surface of the placenta demonstrated a color difference with a plethoric placental share for twin 1 and a pale placental share for twin 2 (Figure 3), in line with the skin color of the infants. The inter-twin membrane appeared thin and translucent. Based on the large inter-twin Hb difference (>8 g/dL) and the presence of only minuscule placental anastomoses, the diagnosis of TAPS was confirmed. At microscopic examination, the inter-twin membrane was found to consist of only amniotic membranes, confirming the twin to be MCDA. The neonatal course of both babies was free from complications. Physical examination showed no abnormalities regarding the external genitalia, and an abdominal ultrasound revealed normal internal genitalia. Karyotyping showed no signs of mosaicism.

## 3. Discussion

These four clinical cases provide valuable insights into various aspects of this rare form of MCDZ twins. The prevalence of this unusual twinning phenomenon remains unclear [7]. Most reported cases of MCDZ twins were initially suspected due to differences in phenotypic sex or ABO blood types observed during cord blood examination. Since many cases go unnoticed, particularly when the twins share the same sex, the prevalence of MCDZ twins might be higher than estimated. Like most cases reported in the literature, all of our cases were identified as MCDZ pregnancies based on sex discordance between the twins, even though the underlying etiology varied for each case.

Chimerism occurs when an organism contains cells derived from more than one distinct zygote [8]. Chimerism may be confined to the blood or may involve non-hematopoietic tissue. While blood chimerism likely results from placental blood-sharing between DZ twins, tissue chimerism is more complex and might arise from ectopic differentiations of chimeric hematopoietic stem cells [9].

Peters at al., in a recent systematic review of 31 MCDZ twins, reported chimerism in 90.3% of all cases, most of which involved only blood chimerism [6]. Tissue chimerism, however is exceedingly rare: only a few cases involving buccal and skin chimerism have been documented [8,9].

The clinical implications of chimerism remain unknown. Only a few cases of atypical genital development have been reported among MCDZ twins with discordant sexes.

Table 1 presents an overview of MC twins with chimerism reported in the literature.

**Table 1 diagnostics-15-00372-t001:** Overview of MC twins showing chimerism reported in the literature.

First Author, Year of Publication	Conception	Pregnancy Complications	Delivery Mode/Wks		Phenotypical Sex		Genital Abnormalities	Chimerism
Nylander, 1970 [10]	Spontaneous				M	F		
Vietor, 2000 [5]			34		M	F	No	Blood
Quintero, 2003 [11]	Spontaneous	TTTS—laser	Miscarriage		M	F	No	Blood
Williams, 2004 [12]	ICSI	Pre-eclampsia	CS	28	M	F	F: clitoromegaly	Blood
Ginberg, 2005 [13]		Expired	SVD	22	M	F	No (autopsy)	Amniotic fluid
Aoki, 2006 [14]	Clomid		CS	34	M	M	No	Blood + blood type
Walker, 2007 [15]	IVF		CS	36	M	M	No	Blood
Ekelund, 2008 [16]	ICSI	TTTS	CS	32	M	F	No	Blood
Hackmon, 2009 [17]	Spontaneous		SVD	37	M	F	No	Blood
Shaikh, 2009 [18]	IVF		CS	34.1	M	F	No	Blood
Bogdanova, 2010 [19]	IUI	Preterm delivery	CS	32	M	F	Female: absence of uterus	Blood
Assaf, 2010 [20]	ICSI	TTTS—laser	CS	37	M	F	No	Blood
Hawcutt, 2011 [21]	IVF	PPROM	CS	25.2	M	F	No	Blood
Loriaux, 2011 [22]		TTTS—IUFT	CS	29.5	M	F	No	Blood
Umstad, 2012 [23]	Spontaneous	TTTS	CS	36	M	M	No	Blood
Choi, 2013 [24]	ICSI	IUFT	CS		M	F	Male: testicular hypoplasia	Blood
Kanda, 2013 [25]	Spontaneous		SVD	36.1	M	F	No	Blood
Smeets, 2013 [26]	IUI		SVD	37	M	F	No	Blood
HJ Lee, 2014 [27]	IVF		SVD	38	M	F	No	Blood
Fumoto, 2014 [8]	IVF	TTTS	CS	35	M	F	No	Blood + buccal
HJ Lee, 2014 [27]	IVF				M	F	No	Blood
Rodriguez-B, 2015 [9]	Spontaneous		CS	39	M	F	M: hypospadias	Blood + skin
Mayeur le bras, 2016 [28]	Clomid	pPROM	CS	36.1	M	F	No	Blood
Gabbett, 2019 [29]	Spontaneous		CS	33	M	F	F: gonadal dysgenesis	Sesquizygotic twins
Suzuki, 2019 [30]	Spontaneous	TAPS						Blood
Chen, 2021 [31]		TAPS	CS	31.2	M	F	No	
Armitage, 2020 [32]	Spontaneous	TTTS	CS	35.2	F	F	No	Blood + Buccal
Daum, 2020 [33]	Spontaneous	TAPS	CS	34	M	F	No	Blood + Buccal
Yoshida, 2021 [34]	Spontaneous		CS	38	M	F	No	Blood+ Buccal+ Blood type
Wimmer, 2022 [35]			CS	36	M	F	M: ambiguous genitalia	Blood + Tetragametic chimerism
Trombetta, 2022 [36]	Spontaneous	Situs Inversus	CS	38	F	F	None	Blood

CS: cesarean section. F: female. ICSI: intracytoplasmic sperm injection. IUFT: intrauterine fetal blood transfusion. IUI: intrauterine insemination. IVF: in vitro fertilization. M: male. pPROM: preterm premature rupture of membranes. TAPS: twin anemia–polycythemia sequence. TTTS: twin-to-twin transfusion syndrome. SVD: spontaneous vaginal delivery.

According to the literature, most MCDZ pregnancies result from assisted reproductive technology (ART) [6]. However, a recent review reported 14 cases of spontaneously conceived MCDZ [36]: both case 1 and case 2 add to the growing body of spontaneously conceived MCDZ twin pregnancies.

Although the precise mechanism underlying MCDZ twinning is uncertain, several hypotheses have been proposed:–The penetration of an oocyte and the second polar body, enclosed by a single zona pellucida, by multiple sperm, leading to distinct genetic contributions [37];–The presence of a binovular follicle with two oocytes surrounded by single zona pellucida that, after being fertilized by different sperm, might merge, resulting in an embryo with genetic material from both oocytes [14];–The fertilization of one ovocyte by two sperm, leading to sesquizygotic multiple pregnancy, a type of twin pregnancy which has intermediate prevalence in both MZ and DZ. In the fertilized egg, there is an independent (heterogonic) assortment of the two paternal genomes and the single maternal genome, leading to a chimeric blastomere, which later undergoes a twinning event: such twins are maternally identical but chimerically share approximately 78% of their paternal genome [29].

While the zygosity test has achieved 99% reliability, the chimerism test should be tailored to each individual case. This requires consultation with a clinical geneticist, as there is variability in its expression in blood and non-hematological tissues.

It is difficult to determine which mechanism led to the MCDZ twin pregnancy after transfer of a single blastocyst in case 3. Multiple pregnancies, including triplets, have been reported after the transfer of a single blastocyst (day five) [38]. However, such cases typically involve MZ twinning. Historically, risk factors for these occurrences have included zygotic cleavage and assisted hatching (laser treatment of the zona pellucida) during ART procedures [39].

The number of MCDZ twin pregnancies reported in the literature is currently insufficient to establish whether the risk of complications is similar or higher compared to that of MCMZ pregnancies.

In our four reported cases, one pregnancy developed TTTS requiring laser therapy: in the literature, so far, only three cases of MCDZ pregnancies which underwent laser surgery for TTTS have been reported [11,20,40]. Of those cases, one underwent a miscarriage after the procedure [11].

Among our four reported cases, only one developed TAPS: according to the literature, three other MCDZ pregnancies involved TAPS development [30,31,33], only in our case, the birth occurred at term via vaginal delivery without complications.

Finally, it is important to consider the hypothesis that, in exceptional and rare cases, TTTS and TAPS may also occur in dichorionic pregnancies [41,42]. Placental histopathological examination is decisive in resolving such a case without attributing it to the phenomena mentioned so far.

## 4. Conclusions

Once considered exceedingly rare, MCDZ twin pregnancies are now gaining increasing recognition and becoming more widely studied. The cases presented in this article provide valuable insights into the complexities of MCDZ pregnancies and highlight their unique characteristics. One key takeaway is the importance of accurately diagnosing chorionicity during the first-trimester ultrasound: proper identification of monochorionicity is crucial for the effective management of potential complications. For MC twins, it is also essential to confirm the concordance of sex of the twins during US: if discordance is detected, amniocentesis is necessary for karyotype evaluation and zygosity testing. The occurrence of blood chimerism adds another layer of complexity to these pregnancies. While blood chimerism appears to be common in MCDZ twins, its clinical implications remain largely unknown. Further research is needed to fully understand the implications of chimerism and its potential impact. Further studies are also needed to evaluate whether the onset of complications related to monochorionicity in MCDZ pregnancies is similar to that in MCMZ pregnancies. As the medical community continues to study this phenomenon, we move closer to providing the best care and support for these pregnancies, emphasizing the importance of exploration and examination of these exceptional occurrences.

## Figures and Tables

**Figure 1 diagnostics-15-00372-f001:**
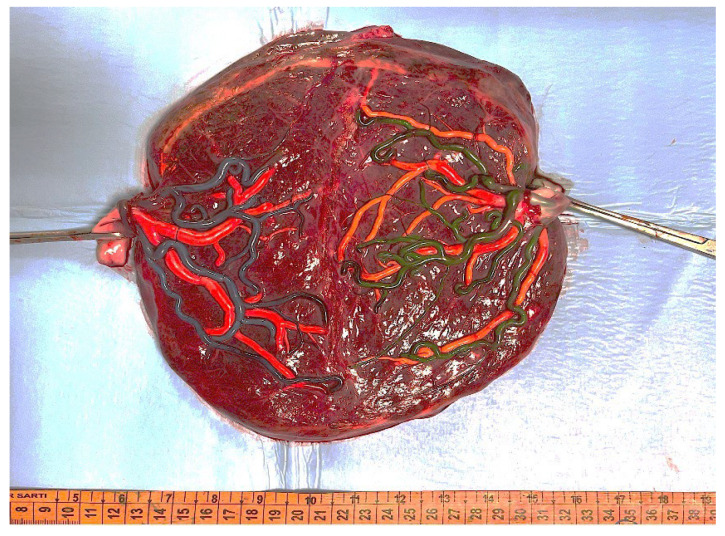
Color dye injection of the placental vessels shows the absence of residual anastomoses after laser surgery.

**Figure 2 diagnostics-15-00372-f002:**
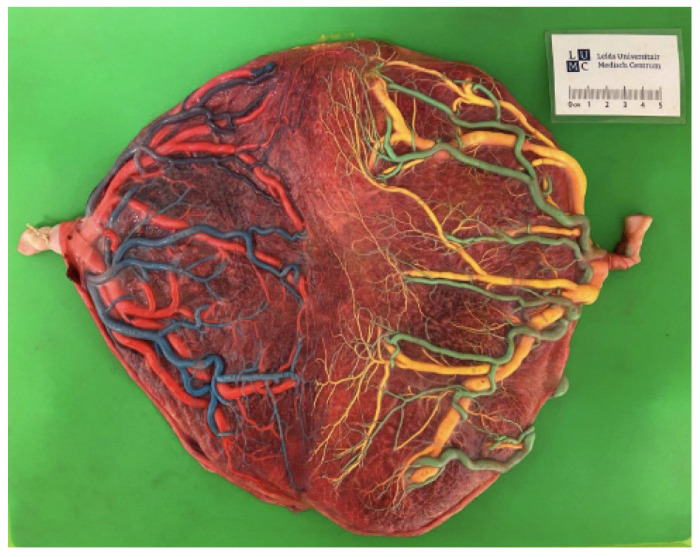
Color dye injection shows multiple vascular anastomoses.

**Figure 3 diagnostics-15-00372-f003:**
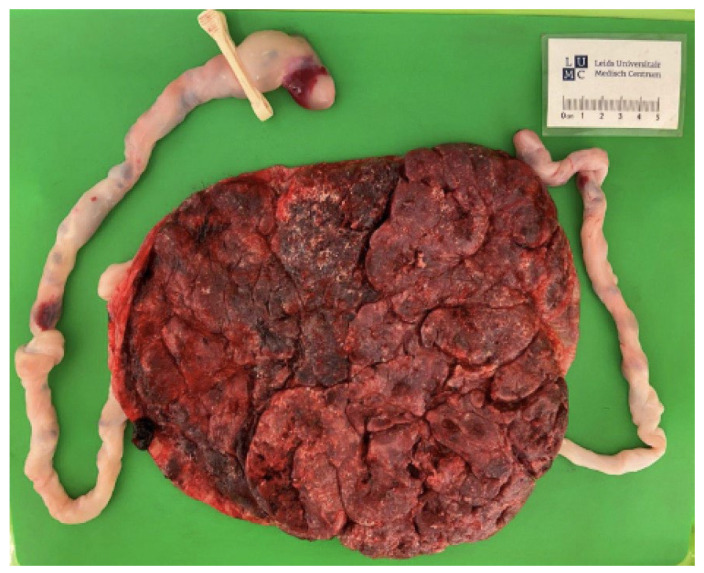
Color difference between the placental shares on the maternal surface of placenta.

## Data Availability

Data are available on request.

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
