# Peer review of "Monochorionic Diamniotic Twins with Sex Discordance: Case Series"

_diagnostics, 2025, doi:10.3390/diagnostics15030372_

Round 1

Reviewer 1 Report

Comments and Suggestions for Authors

Nothing to suggest

Author Response

The paper Case series “Monochorionic diamniotic twins with sex discordance: case series”
Is interesting presenting 4 cases of monochorionic twins with sex discordance, which is a rare
finding
The authors report in the first case, in which at first a normal male karyotype was found, but at 12
weeks a sex discordance was suspected- prob. on the basis of US exam
Case 2 – sex discordance was suspected only after birth (after collecting blood samples for
karyopatype determination, showin 98.5% of male faryotype and 1.5% of female karyotype in the
boy and a normal female kayotype in 48% of the cells and a normal male karyotype in 52% of the
girl - ; non chimerism was found after buccal tissue sampling
Case 3 – MC/DA twinpregnancy after ICSI – susp . TTTS – but without a oligo-polyhydramnios
sequence. He discordance of the sez was based upon amniocenthesis. After the laser procedure x
TTTS cord entanglement between twins was found (iatrogenic monoamnioticity, th sez
discordance was confirmed after birth.
Case 4 – sex of the twint being different, the pregnancy was considered as dichorionic. Finding at
birth of Hb difference – to diagnose the finding as anemia- polycythemia sequence At birth. 1 boy
and 1 girl
The cases are well describe and the discussion is appropriate.
I have nothing to suggest and I proporse to accept the paper like it is. 

Many thanks for your comments and consideration.

Reviewer 2 Report

Comments and Suggestions for Authors

Monochorionic diamniotic twins with sex discordance: case series

Sala et al. reported four twin pregnancies all were pathologically confirmed monochorionic twin gestation with discordant sex by external genital exam and karyotype. This is an interesting entity which is seen rarely in clinical practice and difficult to understand pathophysiology. Authors should be careful with the cases who have sex chromosome mosaicism or chimerism with genital anomaly or ambiguous genitalia. This can be either a different entity or spectrum of the condition that is being discussed in this manuscript.

 I want to congratulate authors for reporting this case series.

After review of the manuscript, these are my suggestions.

1-I suggest authors report the full karyotype of all neonates and their full exam including internal and external genitalia if available. If all pregnancies had zygosity testing this should be included clearly in each case. Is maternal/paternal karyotype included in any of the workup or for investigational purposes?

Authors should include reliability of chimerism and zygosity testing in these situations.

From the reported cases, only single blastocyst transfer has a reliable prediction of monozygotic twin. What is the possibility of twin transfusion complication in fused dizygotic twin pregnancies? Can this be a possible pathophysiology for discordant genotype/sex but TTTS and TAPS?

Discordant phenotype and genotype have been reported and seen in monochorionic twin gestations.  Additionally, chimerism is also reported in monochorionic twin gestations.

2- For case 2- Can you please expand what you mean with “chimerism was not confirmed by buccal swab”? Is chimerism not seen?

3-For Table 1, please use consistent abbreviation (correct to IVF not IFV) and please specify what MPP stands for.

4-Introduction, discussion and conclusions sections are well written. Including different hypothetical explanations is important and helpful.

5- Minor language correction is recommended.

Thank you,

Comments on the Quality of English Language

Manuscript is well written but needs minor revision. Please review.

Thank you,

Author Response

Comment n 1

Sala et al. reported four twin pregnancies all were pathologically confirmed monochorionic twin gestation with discordant sex by external genital exam and karyotype. This is an interesting entity which is seen rarely in clinical practice and difficult to understand pathophysiology. Authors should be careful with the cases who have sex chromosome mosaicism or chimerism with genital anomaly or ambiguous genitalia. This can be either a different entity or spectrum of the condition that is being discussed in this manuscript.
 I want to congratulate authors for reporting this case series.

Response n 1 

Thank you very much for your considerations.

Comment n 2

After review of the manuscript, these are my suggestions.
1-I suggest authors report the full karyotype of all neonates and their full exam including internal and external genitalia if available. If all pregnancies had zygosity testing this should be included clearly in each case.

Response n 2 

Thank you for the suggestions, we have now included a sentence at the end of case 1, 2 and 3 (the 4th already having it) , to accomplish at your request (line 98, line 113, line 140)  .

Comment n 3

 Is maternal/paternal karyotype included in any of the workup or for investigational purposes?

Response n 3

No it was not considered necessary.

Comment n 4

Authors should include reliability of chimerism and zygosity testing in these situations.

Response n 4

Thank you, we have followed your suggestion and included the following sentence in discussion (lines 249-251):
"While the reliability of the zygosity test reaches 99%, the chimerism test should be tailored to each individual case. This requires consultation with a clinical geneticist, as there is variability in its expression in blood and non-hematological tissues."

Comment n 5

From the reported cases, only single blastocyst transfer has a reliable prediction of monozygotic twin. What is the possibility of twin transfusion complication in fused dizygotic twin pregnancies? Can this be a possible pathophysiology for discordant genotype/sex but TTTS and TAPS?

Response n 5

Yes, we have already reported both a case of TTTS and of TAPS in dichorionic twin pregnancy due to fused placentas. We added  in discussion: " Finally, it is important to consider the hypothesis that, in exceptional and rare cases, TTTS and TAPS may also occur in dichorionic pregnancies [22,23]. Placental histopathological examination is therefore decisive in resolving the case without attributing it to the phenomena mentioned so far " .  

Two article are then added in the references 

22.    Lanna M, Faiola S, Casati D, Rustico MA. Twin-twin transfusion syndrome in dichorionic twin pregnancy: rare but not impossible. Ultrasound Obstet Gynecol. 2019 Sep;54(3):417-418. doi: 10.1002/uog.20195. 
23.    Tollenaar LSA, Prins SA, Beuger S, et al. Twin Anemia Polycythemia Sequence in a Dichorionic Twin Pregnancy Leading to Severe Cerebral Injury in the Recipient. Fetal Diagn Ther. 2021;48(4):321-326. doi: 10.1159/000514408. 

Comment n 6

For case 2- Can you please expand what you mean with “chimerism was not confirmed by buccal swab”? Is chimerism not seen? 

Response n 6

Yes the chimerism was not confirmed on buccal swab, so it was considered a blood chimerism due to placental tissue sharing, not involving other tissues(we have modified the sentence at line 111): that is a possibility which is explained in the discussion.

Comment n 7

For Table 1, please use consistent abbreviation (correct to IVF not IFV) and please specify what MPP stands for.

Response n 7

Thanks we have corrected the typo and modified the abbreviation from MPP to preterm delivery.

Comment n 8

Introduction, discussion and conclusions sections are well written. Including different hypothetical explanations is important and helpful.

Response n 8

Many thanks for this comment.

Comment n 9

Minor language correction is recommended.

Response n 9

We have tried to improve the english form , you may find minor corrections throughout the text.